# SALE : Low-bit Estimation for Efficient Sparse Attention in Long-context LLM Prefilling

**Xiaodong Ji** [1]  **Hailin Zhang** [1]  **Fangcheng Fu** [2]  **Bin Cui** [1] [3]

## Abstract

Many advanced Large Language Model (LLM) applications require long-context processing, but the self-attention module becomes a bottleneck during the prefilling stage of inference due to its quadratic time complexity with respect to sequence length. Existing sparse attention methods accelerate attention computation by skipping less significant regions of the attention map. However, these approaches typically perform coarse-grained inspection of the attention map, resulting in their suboptimal performance. In this paper, we propose **SALE**, a fine-grained sparse attention method that accelerates the long-context prefilling stage of LLM with negligible loss in model accuracy. SALE achieves fast and accurate fine-grained attention map estimation using low-bit quantized query-key products to approximate attention weights, followed by the application of a novel *Relative Attention Score* metric to assess the importance of query-key pairs. This design enables us to accurately identify important regions in the attention map, thereby constructing a highly sparse attention mask.

We implement a custom CUDA kernel in SALE optimized for hardware efficiency, reducing overhead to approximately 11% of the full attention latency. Notably, SALE requires no parameter training and can be seamlessly integrated into existing systems with trivial code modifications. Experiments on long-context benchmarks demonstrate that our method outperforms existing approaches in accuracy-efficiency trade-offs, achieving at least 3.36× speedups on Llama-3.1-8B for se-

quences longer than 64K while maintaining model quality. Our code is available at https://github.com/BirdChristopher/SALE

## 1. Introduction

With the growing demand for ultra-long context understanding in complex applications such as long book summarization (Kryściński et al., 2022; Porwal et al., 2023; Chang et al., 2024), long document question-answering (Caciularu et al., 2023; Pang et al., 2022; Fan et al., 2019), and repository-level code completion (Wang et al., 2024a;b), state-of-the-art Large Language Models (LLM) are now equipped with increasingly longer context window (Grattafiori et al., 2024; Yang et al., 2025b; Team et al., 2025; DeepSeek-AI et al., 2025; Xu et al., 2025). Most LLMs employ a decoder-only Transformer architecture (Vaswani et al., 2017), where the self-attention module serves as the core component to enable powerful language understanding capabilities. However, during the prefilling stage of LLM inference, the self-attention module exhibits quadratic time complexity with respect to the number of input tokens. This makes it the primary performance bottleneck, as computational costs increase rapidly with longer contexts (Fu, 2024; Jiang et al., 2024; Lin et al., 2025).

In recent years, numerous research studies have attempted to accelerate prefilling by computing only the important regions of attention maps, based on the observation that attention maps in LLMs are significantly sparse (Deng et al., 2024). These methods, referred to as *sparse attention*, use *sparse masks* to indicate the specific regions of the attention map to be computed. Some sparse attention methods utilize sparse masks with static patterns, such as stride pattern (Child et al., 2019), window pattern (Zaheer et al., 2020; Beltagy et al., 2020), or streaming pattern (Xiao et al., 2024b; Han et al., 2024). However, static sparse masks often result in severe performance degradation, as the real sparse patterns of LLM attention maps are highly dynamic across various input contents (Lai et al., 2025; Jiang et al., 2024; Zhang et al., 2025e).

To adapt to such dynamism, several methods attempt to pre-

[1]School of Computer Science & Beijing Key Laboratory of Software and Hardware Cooperative Artificial Intelligence Systems, Peking University [2]School of Artificial Intelligence, Shanghai Jiao Tong University [3]Institute of Computational Social Science, Peking University(Qingdao). Correspondence to: Bin Cui <bin.cui@pku.edu.cn>.

*Proceedings of the 43rd International Conference on Machine Learning*, Seoul, South Korea. PMLR 306, 2026. Copyright 2026 by the author(s).

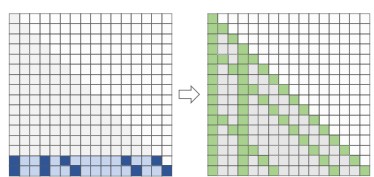 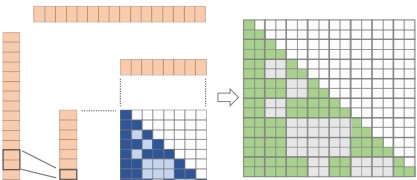 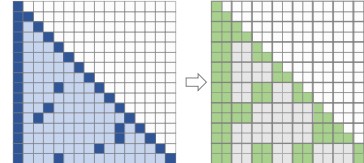

|  (a) Incomprehensive inspection. | (b) Coarse-grained inspection. | (c) Holistic and fine-grained inspection. |

*Figure 1.* Illustration of attention map inspection mechanisms in sparse attention methods. (a) Overview of MInference: It inspects the attention scores of the query tokens at the very end of the sequence to identify important regions. (b) Overview of SpargeAttn: It selects significant regions based on attention scores derived from representative tokens. (c) Overview of SALE: It performs a comprehensive and fine-grained inspection of the entire attention map.

dict critical attention regions by analyzing the attention map. For instance, as illustrated in Figure 1(a), MInference (Jiang et al., 2024) and SampleAttention (Zhu et al., 2024), decompose the sparse attention pattern into combinations of multiple vertical or slash lines, and predict the positions of these lines by analyzing the attention score distribution of a subset of query tokens. Another series of sparse attention methods, such as FlexPrefill (Lai et al., 2025), SpargeAttn (Zhang et al., 2025d), and HiP Attention (Lee et al., 2025), treat the attention map as the concatenation of blocks. As shown in Figure 1(b), they dynamically skip certain attention computations at the block granularity. These methods often construct a representative token for each consecutive query/key chunk and build sparse masks based on the product between representative tokens. Although existing dynamic sparse attention methods can accelerate the prefilling stage of LLM inference to some extent (Ma et al., 2025), they fail to achieve a satisfactory accuracy-efficiency trade-off, as their inspection approaches are either too coarse-grained or insufficiently comprehensive.

In this paper, we propose **SALE**, a novel block-**S**parse **A**ttention technique based on **L**ow-bit **E**stimation of attention weights, to significantly accelerate the long-context prefilling stage of LLM inference with negligible loss in model accuracy. SALE is built on a fast and accurate framework for fine-grained attention map inspection. Figure 1(c) illustrates how SALE inspects the attention map. By performing element-wise importance analysis on the entire attention map, SALE is capable of constructing highly sparse attention masks, while ensuring that the output error is bounded within an acceptable tolerance range. We propose two key components to implement this framework. First, we utilize low-bit quantized query-key products (QK) to approximate attention weights. This process runs efficiently on modern GPUs, leveraging two key factors: the use of high-throughput low-bit Tensor Core instructions and the reduction in global memory access. Second, we propose a novel *Relative Attention Score* metric to evaluate the importance of query-key pairs. Observing that the attention scores in the sink (beginning) and local (end) regions of each attention

map row tend to be relatively higher (Xiao et al., 2024b; Gu et al., 2025), we determine the importance of a query-key pair based on the relative magnitude between its attention weight and the attention weights within the sink and local regions. Compared to common practice that uses the original attention scores (Zhang et al., 2023; Li et al., 2024; Zhang et al., 2025a) as indicators, our design is more efficient because it avoids materializing the entire attention map in DRAM. It also supports adaptive adjustment of the sparsity level according to the input, which allows SALE to robustly handle input samples of varying complexities. SALE further adopts several kernel optimization techniques to optimize hardware efficiency.

SALE incurs no additional training overhead and can be seamlessly integrated into existing inference systems. We conduct comprehensive experiments on various long-context processing benchmarks using two LLMs, Llama-3.1-8B-Instruct (Grattafiori et al., 2024) and Qwen3-4B-Instruct (Yang et al., 2025a), to verify the effectiveness of our method. Experimental results demonstrate that our method delivers a speed-up of at least 3.36× when processing sequences longer than 64K tokens, while maintaining negligible accuracy loss. It achieves superior accuracy-efficiency trade-off compared to the baseline methods.

## 2. Related works

**Sparse LLM inference**    Many previous works try to leverage the sparsity nature of transformer model to accelerate LLM inference from different perspectives. One line of research exploits input text sparsity to dynamically prune context irrelevant to the user's query (Jha et al., 2024; Liu et al., 2025; Shi et al., 2024; Jiang et al., 2023; Li et al., 2023b). While these methods can significantly reduce LLM inference latency for relatively simple prompts, they severely degrade generation quality when processing complex inputs (Yuan et al., 2024).

Numerous studies have observed sparsity patterns in self-attention modules, where only a small subset of attention map elements are much larger than the rest. Some meth-

ods (Xiao et al., 2024b; Fu et al., 2025; Xiao et al., 2024a) use predefined static sparsity patterns to prune the attention map. However, these methods suffer from accuracy degradation as the attention sparsity distribution varies among different input contexts (Jiang et al., 2024; Lai et al., 2025). Other methods assume that the distribution follows certain structures, such as Vertical-Slash or Block-Sparse. Some of them (Jiang et al., 2024; Zhu et al., 2024) try to dynamically predict the location of important regions by examining the exact attention scores of several tokens. Others (Gao et al., 2024; Zhang et al., 2025d; Lai et al., 2025; Lee et al., 2025) regard the attention map of compressed tokens, which are generated from continuous token chunks, as the proxy of real attention map. All these methods fail to achieve accurate predictions due to their overly coarse-grained approximations of attention maps.

In contrast to the aforementioned approaches, several alternatives to self-attention have emerged to circumvent its quadratic complexity. Notable examples include: (1) natively sparse attention algorithms (Yuan et al., 2025; Lu et al., 2025), (2) linear attention mechanisms (Peng et al., 2023; Yang et al., 2024b), and (3) state-space models (Gu & Dao, 2024; Dao & Gu, 2024). However, these methods impose significant adoption costs as they require full model training.

During the decoding stage, methods like SparQ (Ribar et al., 2024) and InfiniGen (Lee et al., 2024) compress the channels of query / key tokens to efficiently approximate the attention scores. Retrieval-based approaches (Zhang et al., 2025a; Chen et al., 2025; Liu et al., 2024) leverage vector-retrieval techniques to approximately sort the attention scores of previous input tokens. Several existing algorithms compress tokens by analyzing attention maps during the prefilling stage. These approaches either eliminate redundant tokens (Zhang et al., 2023; Liu et al., 2023; Li et al., 2024; Ge et al., 2024; Devoto et al., 2024) or perform token merging (Zhang et al., 2024b; Zandieh et al., 2024). Our method is orthogonal to these optimizations and can be combined to further enhance end-to-end LLM inference efficiency.

**Attention kernel optimization** Many CUDA kernel optimization techniques (Dao et al., 2022; Dao, 2024; Shah et al., 2024; Sanovar et al., 2024) leverage hardware features to accelerate the computation of the original full attention. Although these methods accelerate computation, they still require full attention calculations and fail to fully exploit the inherent sparsity of attention maps.

# 3. Method

## 3.1. Problem formulation

We denote the query, key and value matrices as $Q$, $K$ and $V$, respectively, while the corresponding vectors at token offset $i$ are $q_i, k_i, v_i$. Let $N$ represent the sequence length and $d$ represent the hidden size. The shapes of $Q$, $K$ and $V$ are all $N \times d$. Matrix $M$ is the sparse attention mask with a shape of $N \times N$. Single-head self-attention module can be mathematically formalized as below:

$$Attn(Q, K, V, M) = Softmax(\frac{QK^T}{\sqrt{d}} + M) \cdot V \quad (1)$$

During the computation of self-attention, the attention weight matrix $S$ is defined as $S = QK^T/\sqrt{d}$, and the attention score matrix $P$ is defined as $P = Softmax(S + M)$. The sparse attention mask is formed by $M = M_c + M_s$, where $M_c, M_s \in \{0, -\infty\}$ represent the causal mask and the sparse mask respectively. Based on the mathematical properties of Softmax function, if an item $M[i, j]$ in matrix $M$ is $-\infty$, its corresponding attention score will be zero. Therefore, we can skip the attention computation at this position.

For block-sparse attention, query and key tokens are divided into continuous blocks of sizes $b_q, b_k$ along the sequence dimension. We denote the query, key token block at position $j$ as $Q_j, K_j$, which have the shapes of $b_q \times d$ and $b_k \times d$ respectively. For simplicity, we assume $b_q \mid N, b_k \mid N$, and denote $N_q = N/b_q, N_k = N/b_k$. As shown in Figure 2(a), the attention map can be viewed as the concatenation of $N_q \cdot N_k$ attention blocks, each of shape $b_q \times b_k$. Block sparse attention skips certain computations at the block level. To formulate, we denote $M_{bs} \in \{0, 1\}$ as *block-level sparse mask*, and values of sparse mask $M_s$ depend on $M_{bs}$:

$$M_s[i, j] = \begin{cases} 0, & \text{if} \quad M_{bs}[\lfloor i/b_q \rfloor, \lfloor j/b_k \rfloor] = 1, \\ -\infty, & \text{if} \quad M_{bs}[\lfloor i/b_q \rfloor, \lfloor j/b_k \rfloor] = 0 \end{cases} \quad (2)$$

In other words, the attention computation between $Q_i, K_j, V_j$ will be skipped if $M_{bs}[i, j]$ is zero. Block-sparse attention aims to maximize sparsity in matrix $M_{bs}$ while bounding the approximation error relative to full attention within a tolerable threshold.

## 3.2. Block selection via fine-grained importance approximation

Considering the dynamic nature of the attention pattern in LLMs (Jiang et al., 2024; Lai et al., 2025), constructing "Sparse and Accurate" $M_{bs}$ is highly challenging. To achieve this goal, SALE evaluates the "importance" of each position in the attention map. Such a framework enables SALE to examine the attention map at a fine-grained level,

allowing robust handling of various dynamic attention patterns. We propose two key designs—*Low-bit Attention Weight Approximation* and *Relative Importance Approximation*—to ensure the efficient and accurate operation of SALE.

**Low-bit attention weight approximation** SALE first calculates the attention weights for all query-key pairs. Rather than using full-precision floating-point $Q$ and $K$ matrices, SALE computes attention weights using low-bit quantized versions $\widetilde{Q}$ and $\widetilde{K}$ for approximation, significantly minimizing additional overhead with high-throughput low-bit Tensor Core instructions and reduced DRAM access. For instance, on RTX 4090 and RTX 5090 GPUs, the throughput of 4-bit and 8-bit Tensor Core instructions is $8\times$ and $4\times$ that of 16-bit (full precision) Tensor Core instructions, respectively (NVIDIA, 2023; 2025). Meanwhile, the overhead of loading 4-bit and 8-bit QK matrices from DRAM is 1/4 and 1/2 of that required for loading 16-bit QK matrices, respectively.

Regarding the selection of the quantization scheme, we choose to quantize query-key matrices to 4-bit, which maintains high selection accuracy while maximizing the acceleration capability of the low-bit instructions in GPU Tensor Cores. We provide more detailed discussion about quantization scheme in Section I.

**Relative importance approximation** Denoting approximated attention weights as $\widetilde{S}$, the next step is to evaluate the "importance" of each attention block. In related works (Zhang et al., 2023; Li et al., 2024; Liu et al., 2025; 2023), a commonly used metric is the attention score, obtained by applying *Softmax* function to attention weights. Compared to existing practices, our approach performs importance computation on-the-fly, bypassing the need to compute full attention scores. We propose *Relative Attention Score* as our importance metric instead of the original attention scores.

Our design is based on an observation in many related studies (Xiao et al., 2024b; Gu et al., 2025; Xiao et al., 2024a). As shown in Figure 2(b), attention scores within the "sink and local regions" (i.e. the beginning and end of each row) maintain consistently large values, while the region exhibits consistent size across diverse input sequences. Motivated by this pattern, we assess "importance" by comparing $\widetilde{S}[i,j]$ with the attention weights from the sink and local regions. Specifically, before examining those blocks located in the middle of the sequence, we first compute full precision QK matrix multiplication on blocks in the sink and local areas. Denoting the indices of key tokens within the sink and local regions as $I_{SL}$, this process yields two intermediate values, $\widetilde{m}_i$ and $\widetilde{l}_i$. We then compute the *Relative Attention Score* $\widetilde{P}[i,j]$ base on these two intermediate results, which can be

formulated as follows:

$$\widetilde{m}_i = \max_{j \in I_{SL}} S[i,j], \quad \widetilde{l}_i = \sum_{j \in I_{SL}} e^{S[i,j]-\widetilde{m}_i}$$

$$\widetilde{P}[i,j] = \frac{e^{\widetilde{S}[i,j]-\widetilde{m}_i}}{\widetilde{l}_i}$$

If all $\widetilde{P}[i,j]$ values in a block are smaller than the threshold $\tau$ (e.g. 0.004), this block is marked as non-critical, and the attention computation for this block will be skipped. The procedure for determining the threshold value $\tau \in (0,1)$ is elaborated in Section 3.3.

### 3.3. Per-head threshold calibration

Figure 2(b) illustrates the attention score distributions of two attention heads of Llama-3.1-8B-Instruct, exhibiting inconsistent sparsity levels. Thus, applying the same $\tau$ for all heads may lead to suboptimal performance. To address the issue, we propose an offline calibration procedure to determine the optimal $\tau$ value for each head, which ensures negligible output errors while maximizing sparsity.

We adopt the $L_1$ distance between the output of SALE and the output of full attention as the error metric, which can be formulated as $Err(\tau) = \|O - \widetilde{O}\|_1/N$ . $O$ is the result of the original attention, $\widetilde{O}$ is the result of SALE, and $N$ represents sequence length. At the beginning of the calibration, $\tau$ is initially set to be a relatively large threshold $\tau_0$ (e.g. 0.008). We then progressively reduce the sparsity level by halving the value of $\tau$ until $Err(\tau)$ falls below $\theta$, where $\theta$ is the predefined error bound. By tuning $\theta$, we can control the sparsity level of SALE. In Section G, we provide a detailed explanation of the correlations between these hyperparameters.

### 3.4. Kernel optimization

For kernel implementation, we propose several optimization techniques to further improve the hardware efficiency of SALE. Here, we describe some core designs, and more detailed information about the optimizations can be found in Section H.

**Computation tiling** SALE targets long context processing scenarios, where materializing the intermediate results of QK GEMM would occupy a large amount of DRAM space and introduce extensive I/O overhead. Therefore, we implement SALE based on the framework of FlashAttention2 (Dao, 2024) to avoid this issue. Specifically, the computation tasks of different query blocks will be scheduled to different cooperative thread arrays (CTA) for parallel processing. Each CTA iteratively checks the importance between its corresponding query block and different key blocks.

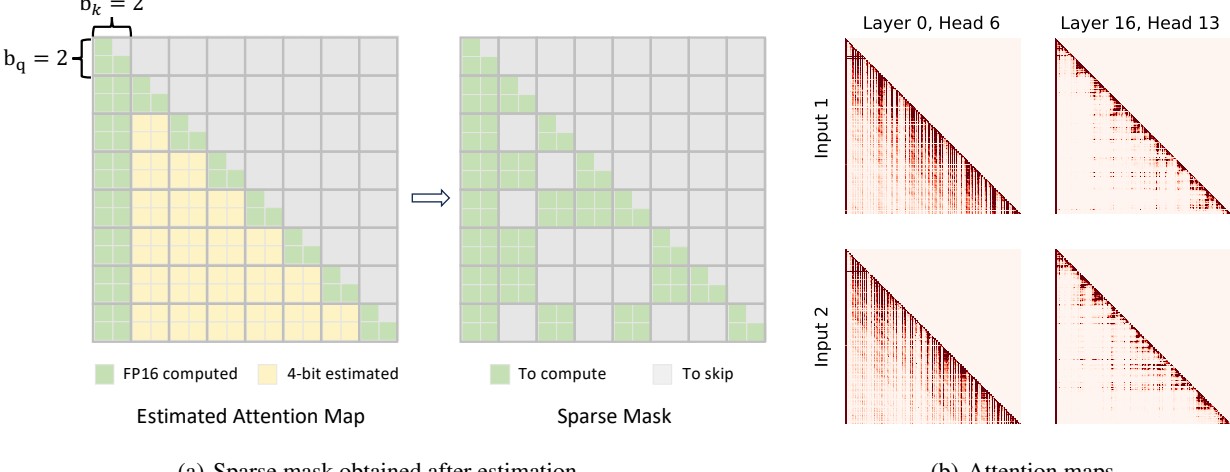

(a) Sparse mask obtained after estimation.

(b) Attention maps.

*Figure 2.* (a) Illustration of SALE. The whole $16 \times 16$ attention map is viewed as concatenation of many $2 \times 2$ blocks. We first estimate the attention weights in an element-wise manner, and then construct a sparse mask at the block level based on these estimations. (b) Attention maps of two different attention heads in Llama-3.1-8B-Instruct when processing different input sequences.

**Estimation-computation disaggregation** In consideration of performance issues, SALE separates "important block selection" and "sparse attention computation" into two distinct CUDA kernels, which are referred to as *Selection-Pass* and *Computation-Pass*, respectively. During Selection-Pass, we select important attention regions at the block granularity and record the coordinates of these blocks. We provide the corresponding pseudo code in Section K for a more detailed description. After Selection-Pass, we compute attention output on selected blocks in the following Computation-Pass.

A potentially simpler and more efficient implementation is the "One-pass" alternative: upon identifying an important key block, immediately compute the output for that block. However, such a design is not feasible in practice, as it fails to reduce computational workload on the one hand and is incompatible with the hardware characteristics of GPUs on the other. We provide a more detailed discussion on this issue in Section J.

**Relative attention score comparison** Directly computing *Relative Attention Score* is time-consuming as it consists of multiple complex hardware instructions, including floating point division and exponential function. Considering that $\widetilde{l}_i$ and $\widetilde{m}_i$ do not change after computation in sink and local areas, we optimize this comparison by following mathematical transformation:

$$\frac{e^{\widetilde{S}[i,j]-\widetilde{m}_i}}{\widetilde{l}_i} \geq \tau \iff \widetilde{S}[i,j] \geq \ln(\tau \cdot \widetilde{l}_i + \widetilde{m}_i) \quad (3)$$

The comparison between the *Relative Attention Score* and $\tau$ can then be accomplished using a single floating point comparison instruction. It is worth noting that we also

mitigate potential overflow issues caused by the exponential function.

**Integration with SageAttention** The final stage of SALE is Computation-Pass. We employ the QK 8-bit quantization strategy proposed in SageAttention (Zhang et al., 2025c) to further accelerate Computation-Pass while maintaining negligible precision loss.

## 4. Experiments

### 4.1. Settings

**Models** Most of the experiments are conducted using Llama-3.1-8B-Instruct (Grattafiori et al., 2024) (**Llama-3.1**). We also use Qwen3-4B-Instruct (Yang et al., 2025a) (**Qwen-3**) to validate the effectiveness of our method on larger-scale LLM.

**Implementation details** We implement Selection-Pass in C++ CUDA and use Triton (Tillet et al., 2019) compiler to accelerate the quantization process (Sun et al., 2025). We implement the quantized Computation-Pass based on the open-source code of SpargeAttn (Zhang et al., 2025d). For those hyper-parameters mentioned in Section 3.2, we use block size $b_q = 64$ and $b_k = 32$. We constrain the sink area size to 32 tokens and the local area size to no more than 256 tokens. During offline calibration, we set the initial threshold $\tau_0 = 0.008$, and use error bounds of $\theta = 0.4$ for Llama-3.1 and $\theta = 4.0$ for Qwen-3 by default. All latency experiments are conducted on a server with 8 GeForce RTX 4090 GPUs without using tensor-parallel (Narayanan et al., 2021; Zhang et al., 2025f) or context-parallel (Li et al., 2023a) techniques. In the Section A, we provide additional

*Table 1.* LongBench evaluation results of different methods. We use **boldface** to denote the highest value and underline to indicate the second-highest value.

| Tasks | Llama-3.1 | | | | | Qwen-3 | | | | |
|---|---|---|---|---|---|---|---|---|---|---|
| | FA2 | MInfer | Flex | Sparge | SALE | FA2 | MInfer | Flex | Sparge | SALE |
| NarrativeQA | **29.93** | 24.92 | 28.29 | 29.62 | 28.95 | **27.99** | 27.63 | 23.82 | 25.29 | 26.67 |
| Qasper | 44.82 | 44.29 | 44.55 | 43.73 | **45.33** | **44.40** | 43.80 | 37.93 | 42.31 | 42.81 |
| MultiFieldQA | 54.65 | 53.71 | 55.34 | **56.02** | 55.18 | **51.49** | 50.30 | 50.30 | 49.61 | 51.17 |
| HotpotQA | 55.81 | 52.00 | 55.38 | 54.57 | **55.83** | **58.30** | 57.72 | 54.62 | 53.17 | 57.15 |
| 2WikiMQA | 46.16 | 44.10 | 43.43 | **47.08** | 42.61 | **42.51** | 41.93 | 35.96 | 35.84 | 42.13 |
| MuSiQue | 30.41 | 25.72 | 30.07 | **31.40** | 30.10 | 22.85 | 22.17 | 19.54 | 23.07 | **23.12** |
| GovReport | 35.29 | 35.09 | 34.64 | 35.04 | **35.45** | **30.61** | 30.56 | 30.00 | 29.22 | 30.42 |
| QMSum | 25.25 | 25.47 | **25.83** | 25.12 | 25.33 | 22.76 | 22.66 | **22.99** | 22.13 | 22.73 |
| TREC | **72.50** | 72.00 | 70.50 | 71.00 | 70.50 | 74.50 | **75.50** | 74.00 | 74.00 | 74.00 |
| TriviaQA | 91.65 | 91.18 | 89.81 | **92.68** | 90.47 | 87.99 | **88.68** | 87.54 | 88.51 | 88.43 |
| SAMSum | 43.67 | 43.73 | 43.18 | 43.18 | **44.19** | 45.35 | 45.47 | 45.25 | **45.62** | 45.51 |
| LSHT | **46.50** | 46.00 | 41.00 | 45.50 | **46.50** | **43.50** | 41.25 | 35.50 | 40.25 | 42.50 |
| Count | 6.72 | 3.25 | 2.59 | 5.89 | **7.09** | 1.33 | 1.08 | 0.00 | 0.67 | **2.58** |
| Retrieval | 99.50 | 97.00 | 82.00 | 84.00 | **100.00** | **100.00** | **100.00** | 56.50 | 91.50 | **100.00** |
| Average | **48.77** | 47.03 | 46.18 | 47.48 | 48.39 | **46.68** | 46.34 | 40.99 | 44.42 | 46.37 |
| Speedup (64K) | 1.00× | 1.07× | 2.21 | 3.11× | **3.36×** | 1.00× | 1.91× | 2.78× | 2.55× | **3.63×** |

*Table 2.* InfiniteBench evaluation results of different methods. We use **boldface** to denote the highest value and underline to indicate the second-highest value.

| Tasks | Llama-3.1 | | | | | Qwen-3 | | | | |
|---|---|---|---|---|---|---|---|---|---|---|
| | FA2 | MInfer | Flex | Sparge | SALE | FA2 | MInfer | Flex | Sparge | SALE |
| Retrieve.KV | 55.60 | 20.00 | 38.00 | 47.20 | **56.40** | 52.00 | 48.00 | **58.00** | 32.00 | 55.00 |
| En.MC | 67.25 | 55.02 | **68.56** | 66.38 | 66.38 | **64.00** | 60.00 | 59.00 | 57.00 | **64.00** |
| Math.Find | 34.29 | **34.86** | 30.00 | 34.57 | 30.57 | 27.00 | 27.00 | **29.00** | 17.00 | 28.00 |
| En.QA | **15.12** | 13.96 | 14.18 | 13.51 | 13.19 | **8.27** | 7.10 | 7.71 | 7.78 | 7.49 |
| En.Dia | 16.50 | 13.50 | 17.00 | 17.50 | **19.00** | **8.00** | **8.00** | 6.00 | 7.00 | 6.00 |
| Average | **37.75** | 27.47 | 33.55 | 35.83 | 37.11 | 31.85 | 30.02 | 31.94 | 24.14 | **32.09** |
| Speedup (64K) | 1.00× | 1.07× | 2.21 | 3.11× | **3.36×** | 1.00× | 1.91× | 2.78× | 2.55× | **3.63×** |

information about the implementation.

**Baselines** To demonstrate the advantages of SALE, we compare it with four strong baselines for self-attention acceleration in long-context processing: **FlashAttention2**(*FA2*) (Dao, 2024), **MInference**(*MInfer*) (Jiang et al., 2024), **FlexPrefill**(*Flex*) (Lai et al., 2025), and **SpargeAttn**(*Sparge*) (Zhang et al., 2025d). FA2 computes standard full attention, which serves as an oracle. The other three methods employ sparse attention mechanisms. All experimental results are based on their publicly available implementation. We use $\gamma = 0.95$ for Llama-3.1 and $\gamma = 0.93$ for Qwen-3 when evaluating FlexPrefill. We use $(l_1 = 0.08, l_2 = 0.09)$ for both Llama-3.1 and Qwen-3 when evaluating SpargeAttn. For MInference, we select the optimal sparse pattern based on its open-source code. Additionally, to investigate the performance of these methods under varying sparsity levels, we prepare multiple sets of hyperparameters based on their publicly available codes. For FlexPrefill and SpargeAttn, as described in their papers, we adjust their sparsity levels by tuning $\gamma$ and $(l_1, l_2)$, respectively. For MInference, since its open-source implementation configures all heads with the *Vertical-Slash* pattern,

the sparsity rate is adjusted by varying the total number of vertical and slash lines across all heads. We provide more details in Section B.

**Metrics** To validate the effectiveness of SALE, we assess model quality using long-context benchmarks (see Section 4.2) and measure efficiency through latency measurements. All latency results in the experimental section focus solely on the attention computation time across all layers during the LLM prefilling phase. Our latency measurements include all online operations, such as quantization, block selection, and index selection. In some experiments, we report the end-to-end (E2E) latency on certain datasets, which is computed by summing the latency of all samples in the dataset.

### 4.2. Accuracy evaluation

Following common practice (Zhang et al., 2025d; Jiang et al., 2024; Lai et al., 2025; Zhang et al., 2025a; Gao et al., 2024; Li et al., 2024), we adopt three long-context understanding benchmarks to compare the generation quality of our method with other baselines. Achieving higher scores

*Table 3.* RULER evaluation results of different methods. We use **boldface** to denote the highest value and underline to indicate the second-highest value.

| Context length | Llama-3.1 | | | | | Qwen-3 | | | | |
|---|---|---|---|---|---|---|---|---|---|---|
| | FA2 | MInfer | Flex | Sparge | SALE | FA2 | MInfer | Flex | Sparge | SALE |
| 8K | 92.10 | 92.52 | 91.44 | **92.68** | 91.68 | 95.46 | 95.95 | 92.77 | 71.06 | **96.04** |
| 16K | 89.56 | 87.41 | **90.54** | 89.83 | 89.51 | 94.27 | **95.09** | 93.49 | 81.99 | 93.62 |
| 32K | 84.43 | 84.17 | **85.74** | 84.15 | 85.22 | **93.17** | 91.42 | 91.63 | 81.33 | 92.68 |
| 64K | 80.81 | 81.54 | 80.46 | 80.03 | **82.12** | 90.24 | 87.26 | 88.68 | 86.22 | 90.10 |
| 128K | 72.43 | 61.41 | 72.57 | 71.91 | **74.81** | **83.82** | 82.17 | 80.46 | 80.45 | 83.47 |
| Average | 83.86 | 81.41 | 84.15 | 83.72 | **84.67** | **91.39** | 90.38 | 89.41 | 80.21 | 91.18 |
| Speedup (64K) | 1.00× | 1.07× | 2.21 | 3.11× | **3.36×** | 1.00× | 1.91× | 2.78× | 2.55× | **3.63×** |

on these baselines indicates better performance. (1) **Long-Bench** (Bai et al., 2024): A comprehensive benchmark covering diverse long-text applications, including single-document QA, multi-document QA, summarization, few-shot learning, synthetic tasks, etc. The context lengths of most input samples are below 32K tokens. (2) **InfiniteBench** (Zhang et al., 2024a): A benchmark designed to evaluate the capability of processing excessively long context (exceeding 100K tokens). It comprises several challenging synthetic tasks such as Retrieve.KV and Math.Find, as well as other real-world tasks including QA and summarization based on fake books or fake dialogues. (3) **RULER** (Hsieh et al., 2024): A widely used synthetic long-context benchmark, including tasks such as Needle-In-A-Haystack and Multi-hop Tracing. Both task difficulty and context length can be flexibly configured.

**LongBench** Table 1 presents the LongBench evaluation results comparing SALE with baseline approaches. In the second row of the table, we use abbreviations introduced in Section 4.1 to denote each method. In the last two rows, we report the average scores and the latency speedup. The latency metric, measured on a 64K-length Needle-In-A-Haystack input, confirms the premise of our accuracy test, that SALE achieves the highest speedup when processing long contexts.

The results on both models show that SALE achieves superior accuracy among all sparse attention baselines.

**InfiniteBench** Table 2 presents the test scores of InfiniteBench, evaluating the capability of processing extremely long inputs. As shown in the table, our method also achieves the best accuracy-efficiency trade-off on InfiniteBench. In addition, when our method is applied, Llama-3.1 exhibits only marginal performance degradation while Qwen-3 even shows improvement. We attribute this improvement to our method's ability to potentially filter noisy information during the prefilling phase, thereby enhancing the model's comprehension capabilities.

**RULER** Table 3 presents the evaluation results on the RULER benchmark. We compare our method against various sparse attention baselines across different context lengths and report the average score for each approach. As shown in the table, our method consistently outperforms other sparse attention baselines in terms of both efficiency and accuracy.

### 4.3. Efficiency evaluation

**Single input speedup** We first compare the latency of different methods when processing a single input. The results are presented in Figure 3(a). We conduct experiments using Llama-3.1 and report the speedup of each method relative to FlashAttention2. To illustrate how latency scales with the number of tokens, we prepare four input samples of different lengths. These samples are obtained by truncating a single 128K-length input from the Needle-In-A-Haystack task in RULER.

Our method demonstrates consistent speedups over FlashAttention2 across all sequence lengths while outperforming all sparse attention baselines in most cases. At 16K context length, SALE achieves a 1.99× speedup. It further exhibits greater speedup as context length increases—reaching 3.88× at 128K—owing to sparser attention patterns. It should be noted that the hyperparameters of each sparse method in this test are aligned with those in Section 4.2, indicating that our speedup evaluation is based on SALE achieving optimal accuracy performance.

**Accuracy vs efficiency** We adjust the computation budget of each method following the approach described in Section 4.1 to analyze the accuracy-efficiency trade-offs. Considering that the speedup achieved by dynamic sparse attention methods may vary depending on the input content, we evaluate the end-to-end latency of all methods on LongBench, InfiniteBench and RULER for comprehensive comparison. The results, shown in Figure 4, demonstrate the superior performance of our method on these datasets.

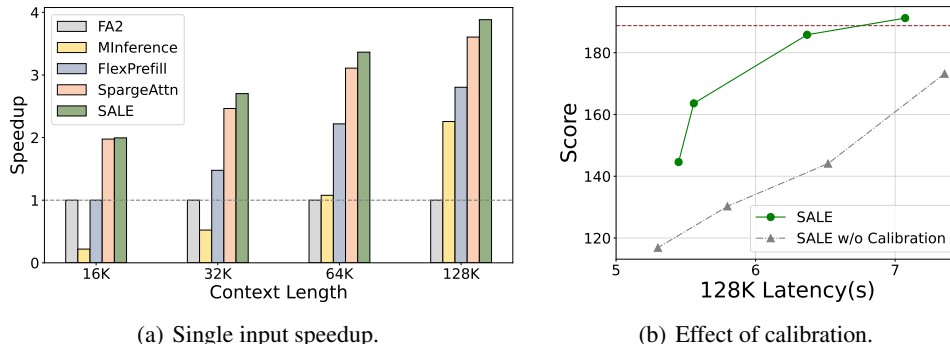

(a) Single input speedup.

(b) Effect of calibration.

*Figure 3.* (a) Speedup in single-input processing. (b) Comparison between SALE v.s. SALE w/o Calibration on InfiniteBench. The brown horizontal dashed line represents the score achieved by FlashAttention2.

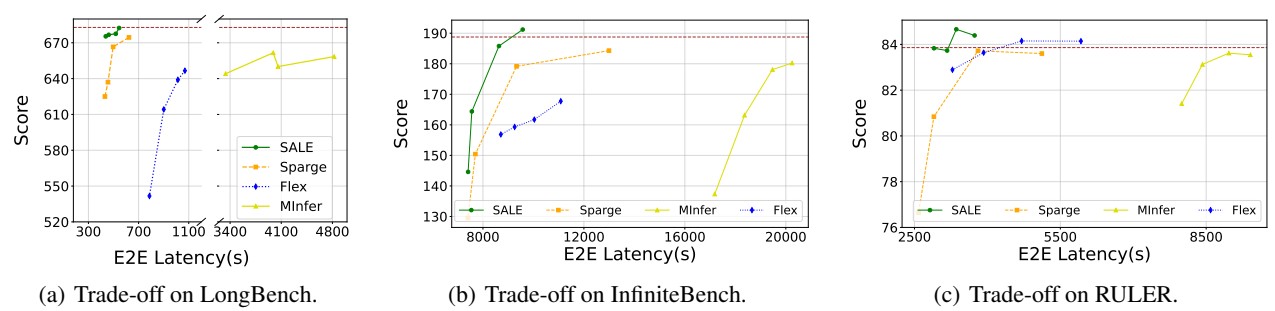

(a) Trade-off on LongBench.

(b) Trade-off on InfiniteBench.

(c) Trade-off on RULER.

*Figure 4.* Evaluation of accuracy-efficiency trade-offs. The brown horizontal dashed line represents the score achieved by FlashAttention2. Tradeoff curves demonstrates that SALE achieves the best speed-accuracy trade-off across all three datasets.

## 4.4. Ablation study

In this section, we evaluate the latency of each stage in SALE and assess the impact of per-head threshold calibration. Additionally, in Section D and Section I, we provide additional experimental results to demonstrate the block selection accuracy of SALE. We also present sparsity measurement results of these sparse methods across contexts of varying lengths in Section F, which serves as a supplement to our previous comparison.

**Latency breakdown** We report the latency breakdown results of SALE under various input lengths in Table 4. All experiments use Llama-3.1, with reported timings reflecting end-to-end execution across all 32 model layers. In the second-to-last line, we show the execution time ratio of Quantization and Selection-Pass operations relative to full attention latency. In the final line, we present the speedup of Computation-Pass compared to full attention. The results demonstrate that our method introduces acceptable computational overhead, with its relative cost decreasing as sequence length grows. Furthermore, Computation-Pass shows greater speedups with longer context lengths, reflecting improved sparsity level at scale.

**Threshold calibration** To demonstrate the performance gain brought by per-head threshold calibration, we set all

*Table 4.* Latency breakdown (ms).

| Context length | 8K | 16K | 32K | 64K | 128K |
|---|---|---|---|---|---|
| Quantization | 11 | 21 | 47 | 99 | 208 |
| Selection-Pass | 14 | 48 | 166 | 634 | 2562 |
| Computation-Pass | 51 | 137 | 378 | 1117 | 3599 |
| FA2 | 106 | 416 | 1597 | 6224 | 24731 |
| Overhead ratio | 23.9% | 16.7% | 13.3% | 11.5% | 11.1% |
| Computation-Pass speedup | 2.08× | 3.04× | 4.23× | 5.57× | 6.87× |

heads in Llama-3.1 to share the same $\tau$, which is referred to as *SALE w/o Calibration*. As shown in Figure 3(b), setting different $\tau$ values according to the characteristics of attention patterns across different heads yields substantial performance gains.

## 5. Conclusion

In this paper, we propose a block-**S**parse **A**ttention technique based on **L**ow-bit **E**stimation. By performing fine-grained estimation of the attention map, we achieve a better accuracy-efficiency trade-off. Specifically, we estimate the attention weights using low-bit quantized queries and keys, and assess the importance of query-key pairs using our *Relative Attention Score* metric. Furthermore, we introduce several CUDA kernel optimization techniques to ensure the

efficiency of sparse mask construction on hardware. Experimental results demonstrate that our approach achieves the best trade-off among existing sparse attention baselines, delivering a speedup of at least $3.36\times$ when processing sequences longer than 64K tokens while maintaining negligible accuracy loss.

## Limitation

This work targets acceleration of the prefill phase for long-context LLM inference, but it shows limited effectiveness on two workload categories. When input sequences are shorter than roughly 8K tokens, attention computation itself incurs little cost. The additional overhead from low-bit estimation exceeds the savings achieved by block skipping, resulting in no tangible end-to-end advantage compared with full attention. For decoding-bound workloads characterized by short prompts and lengthy generation, the prefill stage accounts for minimal overall runtime, and its optimization cannot mitigate user-perceived latency. We note that decoding-stage acceleration is an orthogonal research direction and is not covered in this paper.

Due to our method's reliance on high-throughput 4-bit Tensor Core instructions to accelerate the Selection-Pass, it may lose its performance advantage on hardware that does not support efficient 4-bit matrix multiplication. Moreover, our current implementation is limited to approximating attention weights using Int4 quantization. Additional adaptations would be needed to deploy our method on hardware that supports FP4 GEMM or LUT-based low-bit GEMM. We leave it as our future work.

## Acknowledgements

This work is supported by National Natural Science Foundation of China (U23B2048, 62402011), Fundamental and Interdisciplinary Disciplines Breakthrough Plan of the Ministry of Education of China (JYB2025XDXM108), and High-performance Computing Platform of Peking University.

## Impact Statement

This paper presents work whose goal is to advance the field of machine learning. There are many potential societal consequences of our work, none of which we feel must be specifically highlighted here.

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

# A. Additional implementation details

For the quantization algorithm scheme, we use the smoothing technique mentioned in SageAttention2 (Zhang et al., 2025b) to improve the quality of QK quantization, which introduce negligible overhead. While the quality of quantization affects the accuracy performance of SALE, it should be noted that SALE is designed to be orthogonal to the quantization algorithm.

We select five input samples from the Retrieve.KV task in InfiniteBench to perform calibration for SALE, and the final configuration must satisfy the error bound requirement across all five samples. The per-head threshold calibration for Llama-3.1 on RTX4090 server takes approximately five minutes to complete.

# B. Additional experiment details

For model inference, we leverage the transformers (Wolf et al., 2020) library to build an execution pipeline and replace the default self-attention module with sparse methods. We use greedy decoding to avoid randomness during generation, and use the default chat template to construct the input prompt. We use the same set of input samples as our method to search the optimal hyperparameters for SpargeAttention. Since the open-source code of MInference only uses one input sample for calibration, we employed the first sample from this set for its sparse pattern searching.

During evaluation process, to ensure proper model behavior, we truncate samples that exceed the maximum context window length. Following common practice, we retain the tokens from both the beginning and the end of the sequence and remove those from the middle portion. For all these benchmarks and tasks, we employ the official evaluation scripts from their respective open-source repositories to assess model outputs.

For the data format during model inference, we employed BFloat16 for FlexPrefill due to requirements specified in its repository, while Float16 was used for all other methods.

# C. Performance comparison on larger model

To provide a more comprehensive comparison with other methods, we evaluated the performance of each approach based on the Qwen-2.5-32B-Instruct model (Yang et al., 2024a). Table 5 and Table 6 demonstrate the evaluation results. Evaluation results show that SALE's advantages persist even with larger models.

*Table 5.* LongBench evaluation results of different methods. We use **boldface** to denote the highest value and underline to indicate the second-highest value.

| Tasks | Qwen-2.5 | | | | |
|---|---|---|---|---|---|
| | FA2 | MInfer | Flex | Sparge | SALE |
| NarrativeQA | 29.20 | 31.27 | 29.80 | 29.19 | **32.21** |
| Qasper | 45.79 | 45.05 | 45.53 | 44.61 | **45.95** |
| MultiFieldQA | 53.25 | 53.01 | 52.61 | 51.66 | **53.37** |
| HotpotQA | 64.68 | 64.59 | **64.78** | 63.94 | 63.95 |
| 2WikiMQA | 60.87 | 60.82 | **62.98** | 61.13 | 62.33 |
| MuSiQue | 39.89 | **41.38** | 39.46 | 39.22 | 40.54 |
| GovReport | 30.38 | 30.59 | **30.78** | 30.36 | 30.66 |
| QMSum | 23.06 | 23.16 | 23.10 | 23.18 | **23.42** |
| TREC | 73.50 | 73.50 | 73.50 | **74.50** | 73.00 |
| TriviaQA | 87.68 | 88.40 | **89.40** | 88.81 | 87.97 |
| SAMSum | 45.67 | 45.92 | **46.43** | 46.41 | 45.92 |
| LSHT | 45.79 | **47.50** | 44.17 | 47.00 | 47.21 |
| Count | 12.67 | **13.67** | 3.57 | 9.22 | 13.38 |
| Retrieval | **99.50** | 99.25 | 92.25 | 98.83 | 98.25 |
| Average | 50.85 | 51.29 | 49.88 | 50.57 | **51.30** |
| Speedup (64K) | 1.00× | 1.25× | 1.39× | 2.55× | **3.28×** |

# D. 4bit Selection-Pass vs 16bit Selection Pass

To evaluate the effectiveness of 4-bit attention weight approximation, we further conducted experiments using original-precision (16-bit) QK matrices to inspect the attention map, which is referred to as *SALE w/o QK Quant*. The result is shown in Figure 5. We measure the single input speedup of two methods under varying input lengths, using the same set of input

*Table 6.* InfiniteBench evaluation results of different methods. We use **boldface** to denote the highest value and underline to indicate the second-highest value.

| Tasks | Qwen-2.5 | | | | |
| --- | --- | --- | --- | --- | --- |
| | FA2 | MInfer | Flex | Sparge | SALE |
| Retrieve.KV | 4.00 | **7.60** | 4.80 | 4.60 | 5.40 |
| En.MC | **63.70** | 63.32 | 59.80 | 61.50 | 62.80 |
| Math.Find | 41.40 | 45.71 | 47.20 | 41.70 | **52.00** |
| En.QA | 6.70 | 6.85 | 6.70 | 6.80 | **6.90** |
| En.Dia | 27.50 | **30.00** | 29.00 | 25.50 | 27.50 |
| Average | 28.66 | 30.70 | 29.5 | 28.02 | **30.92** |
| Speedup (64K) | 1.00× | 1.25× | 1.39× | 2.55× | **3.28×** |

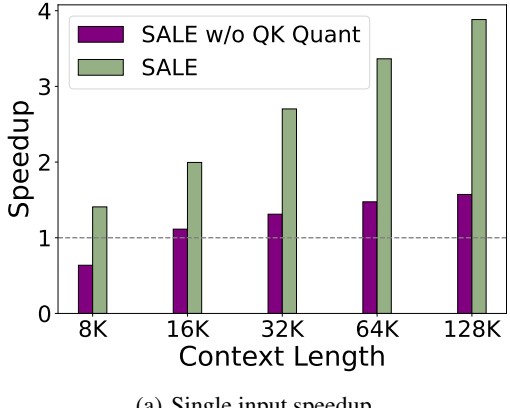

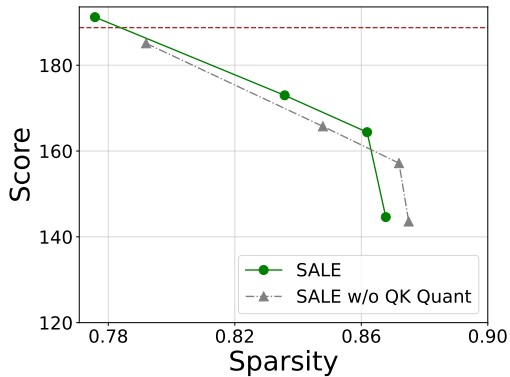

(a) Single input speedup        (b) Accuracy and sparsity result

*Figure 5.* Comparison between SALE and SALE w/o QK Quant. (a)Single input speedup. (b) Comparison between SALE v.s. SALE w/o QK Quant on InfiniteBench. The brown horizontal dashed line represents the score achieved by FlashAttention2.

samples as in the Figure 3(a). The result indicates that using original-precision QK to estimate attention weights leads to a significant increase in computational overhead.

We further evaluate the accuracy and attention sparsity of both methods based on Llama-3.1, where corresponding data points for the two methods are obtained using the same $\theta$. We use the scores from InfiniteBench to represent accuracy. Attention sparsity metric is defined as the ratio of the number of skipped attention blocks to the total number of attention blocks, and the results presented here are measured when processing contexts of 128K length. As observed, under identical hyperparameter settings, *SALE w/o QK Quant* achieves higher attention sparsity while showing a slight performance drop on InfiniteBench. This may be attributed to the limited precision of current Int4 quantization techniques, which can cause certain approximated attention weights to exceed their true values, thereby leading to more blocks being selected.

# E. Latency comparison with quantized full attention

To clarify how much of SALE's speedup comes from sparsity rather than from low-bit attention itself, we compare SALE against SageAttention (Zhang et al., 2025c), an INT8-quantized full-attention scheme. We measure single-input attention latency on Llama-3.1-8B-Instruct using a single RTX 4090, sweeping context length from 8K to 128K. SALE uses hyperparameter $\theta = 0.4$ throughout.

*Table 7.* Attention latency (ms) comparison

| Method | 8K | 16K | 32K | 64K | 128K |
| --- | --- | --- | --- | --- | --- |
| Full (FP16) | 107 | 423 | 1642 | 6404 | 25657 |
| Sage (INT8) | 78 | 285 | 1076 | 3994 | 15592 |
| SALE ($\theta = 0.4$) | **73** | **203** | **592** | **1876** | **6524** |
| Speedup over Sage | 1.06× | 1.40× | 1.82× | 2.13× | 2.39× |

Although SageAttention already provides a substantial speedup over FlashAttention2, SALE delivers an additional $1.06\times-2.39\times$ speedup on top, with the gap widening as context grows. This confirms that block skipping based on low-bit estimation is complementary to attention quantization. Even with quantized dense kernels, sparsity exploitation yields considerable further gains in long-context settings.

## F. Sparsity statistic analysis

To enable a more transparent comparison between SALE and other sparse attention baselines introduced in the Section 4, we tested the sparsity results of these methods on the input samples from LongBench. We divided all inputs into three groups according to their context lengths and computed the average sparsity rate along with its standard deviation for each group during processing. The hyperparameters used for each sparse method here are consistent with those described in Section 4.1. The results are in Table 8, with data in each cell formatted as (mean, std):

*Table 8.* Sparsity comparison

| Sparse methods | MInfer | Flex | Sparge | SALE |
|:---:|:---:|:---:|:---:|:---:|
| 0-20K | (0.138, 0.029) | (0.604, 0.035) | (0.184, 0.016) | (0.517, 0.028) |
| 20K-40K | (0.356, 0.043) | (0.676, 0.046) | (0.279, 0.022) | (0.651, 0.028) |
| > 40K | (0.471, 0.044) | (0.677, 0.026) | (0.339, 0.025) | (0.678, 0.020) |

The experimental results show that our scheme has obvious advantages in terms of sparsity rate compared with MInference and SpargeAttn. When compared with FlexPrefill, our sparsity rate is lower for context length $\leq$40K, but higher for context length > 40K.

It should be noted that a higher sparsity rate of FlexPrefill here does not mean better performance. According to the results in Figure 4(a), SALE can achieve both higher scores and faster end-to-end latency than FlexPrefill under various sparsity rates.

## G. Hyperparameters explanation

$\tau$ controls the sparsity rate of a specific attention head. When the relative attention score of an attention block is less than $\tau$, this block will be ignored in the Computation Pass. Therefore, as $\tau$ increases, more blocks will be skipped, leading to a higher sparsity rate, decreased accuracy, and faster attention computation. The value of $\tau$ can vary across different heads, and its specific value is determined during the calibration process.

$\tau_0$ represents the initial value of $\tau$ in calibration process for each attention head. It will gradually decrease during the calibration process until the output error requirement is met. Therefore, it is sufficient for $\tau_0$ to have a relatively large initial value. $\tau_0$ does not affect the speedup and accuracy performance of the model.

$\theta$, on the other hand, is a hyperparameter that adjusts the global sparsity rate. When $\theta$ increases, the $\tau$ values for each head will also increase, and the sparsity level of SALE get higher. In our Figure 4, we have assessed the performance of SALE under various values of $\theta$.

## H. Additional Kernel optimization technique

**Reduction in dequantization operations**  Theoretically, whether an attention block is skipped only depends on the comparison between the largest *Relative Attention Score* with $\tau$. By employing per-thread quantization strategy proposed in (Zhang et al., 2025b), we make all quantized attention weight elements held by each thread share the same quantization scale. This ensures that the largest *Relative Attention Score* and the largest approximated attention weight occur at the same position. Therefore, only the largest approximated attention weight needs to be dequantized, which saves many low-throughput operations such as datatype conversion.

**Segment level all-reduce**  Due to the hardware characteristics of GPU Tensor Cores, the QK GEMM for an attention block is collectively executed by multiple threads within a GPU thread blocks with the output of the QK GEMM stored in a distributed manner across different threads. Since the elements held by each thread are invisible to other threads, we need to perform a CTA-wise all-reduce operation on the relative importance comparison results in each thread. This ensures that all

threads within the thread blocks reach a unified judgment on the importance of the current block.

In fact, this all-reduce operation is costly:it introduces multiple `warp_shfl` instructions, `_syncthreads` instructions, and shared memory access instructions. Since the Selection-Pass does not require computing attention output, we employs specific optimization techniques to reduce the number of all-reduce operations. Specifically, each thread can temporarily store the relative importance of multiple consecutive blocks as multiple bits in local variables, and use a single all-reduce operation to achieve CTA-wise consistency. This allows us to save a significant number of all-reduce operations.

## I. Quantization scheme analysis

SALE uses quantized QK to estimate attention weights in Selection-Pass. Theoretically, the fewer the number of bits used in quantization, the faster the estimation of attention weights will be, as this allows the use of faster Tensor Core instructions and results in lower DRAM memory access overhead. Considering that the minimum bit width supported by Tensor Cores in current mainstream computing cards is 4-bit, the most efficient quantization scheme we can choose is 4-bit quantization for QK. Lower quantization bit widths cannot yield additional performance benefits.

However, lower-precision quantization also introduces greater errors, possibly leading to a decrease in the accuracy of important block selection in Selection-Pass. To demonstrate the block selection accuracy under 4-bit quantization, We tested the "recall rate" between the blocks selected using 4-bit QK and those selected using full precision floating-point QK. The "recall rate" metric can be defined as "the number of blocks commonly selected by both methods" divided by "the total number of blocks selected using full precision QK". The input samples used in the test are aligned with those in Figure 3(a), and the tests are also based on the Llama-3.1-8B-Instruct model. We obtained the statistical results under different context lengths, which are demonstrated in Table 9:

*Table 9.* Block selection recall rate

| Context length | 8K | 16K | 32K | 64K | 128K |
|:---:|:---:|:---:|:---:|:---:|:---:|
| Recall rate | 98.2% | 97.7% | 97.2% | 96.6% | 95.4% |

The experimental results demonstrate that 4-bit QK is capable of accurately selecting important blocks. The "Sparsity-Accuracy" curve presented in Figure 5(b) also indicates that quantizing QK to 4-bit can achieve block selection accuracy comparable to that of full-precision QK. Thus, SALE adopts 4-bit scheme which enables it to balance accuracy and efficiency.

## J. Two pass design rationale

In contrast to our "Estimation-computation disaggregation" ("Two-pass") kernel design mentioned in Section 3.4, the "One-pass" design integrates these two processes into a single kernel. Specifically, when iterating through a key block, if the current attention block is determined to be important, the output for the current block is computed directly (i.e. "on-the-fly pruning").

Theoretically, this design eliminates the need to store the coordinates of important blocks, and can reuse the computation results of low-bit QK GEMM, which is expected to run faster. However, in practice, the "One-pass" design is not feasible. Two primary reasons are listed here:

**4-bit quantization issue** In fact, the results of 4-bit QK GEMM cannot be directly used for computing attention output, as this would lead to significant accuracy degradation. To illustrate this point, in Table 10, we present the accuracy experiment results of using 4-bit QK in the computation stage. We refer to this scheme as "4+4", whereas SALE corresponds to "4+8". We also tested a scheme that calculates attention scores using only 4-bit QK without block pruning, denoted as "4 dense". All 4-bit quantization in the table has adopted the technique mentioned in SageAttention2 (Zhang et al., 2025b) to improve quantization quality. We report the scores on two tasks of InfiniteBench in Table 10:

Our experimental results show that the "4+4" scheme leads to a degradation in model performance. Even if we do not prune any blocks, a significant score drop still occurs. Therefore, we choose to use QK with a precision of at least 8 bits in the Computation Pass.

*Table 10.* Accuracy performance comparison

| Tasks | Retrieve.KV | En.MC |
|---|---|---|
| 4+4 | 35.8 | 63.30 |
| 4 dense | 41.4 | 60.26 |
| SALE (4+8) | 56.4 | 66.38 |

**Implementation challenges** According to the hardware characteristics of GPUs, "on-the-fly pruning" cannot be efficiently implemented on GPUs. The requirement for immediate decisions imposed by the "on-the-fly pruning" means that we have to perform an all-reduce operation each time after computing the QK GEMM. Based on the analysis provided in Section H, it will introduce much more CTA-wise all-reduce operations compared to SALE, which is not efficient.

In addition, "on-the-fly pruning" is unfriendly to the GPU memory hierarchy. Due to the high DRAM access latency of GPUs, high-performance CUDA implementations typically pre-issue memory access instructions for data that will be computed to achieve overlap between memory access and computation. However, the nature of "on-the-fly pruning" dictates that we must decide whether to issue the memory access instruction only after completing the QK GEMM, and then wait for the data transmission of value tensor, which waste the computational resource on SM. Otherwise, if we issue memory access instructions for the value tensor for all blocks, the unnecessary memory access will also impair performance.

Based on the above analysis, even if we adopt the "8+8" scheme to attempt reusing the results of 8-bit QK GEMM, it still cannot match the efficiency of "Two-pass" design.

## K. Selection-Pass algorithm

---
**Algorithm 1** Selection-Pass

---
**Input:** $Q, K \in \mathbb{R}^{N \times d}$, 4-bit quantized matrices $\widetilde{Q}, \widetilde{K} \in \mathbb{Z}^{N \times d}$, threshold $\tau$, block size $b_q, b_k$, local area size $l$.

1   $N_q \leftarrow N/b_q, N_k \leftarrow N/b_k, N_{local} \leftarrow l/b_k$   Split $Q, K$ into blocks $Q_i \in \mathbb{R}^{b_q \times d}, K_j \in \mathbb{R}^{b_k \times d}$, split $\widetilde{Q}, \widetilde{K}$ into blocks $\widetilde{Q}_i \in \mathbb{Z}^{b_q \times d}$, $\widetilde{K}_j \in \mathbb{Z}^{b_k \times d}$ **for** $i = 0$ *to* $N_q - 1$ **do**

2      $I_{SL} \leftarrow \{0\} \cup [i - N_{local}, i - 1]$;                 // Block indices of sink-local area

3      $\widetilde{m}, \widetilde{l} \in \mathbb{R}^{b_q}, \widetilde{m} \leftarrow -\infty, \widetilde{l} \leftarrow 0$;               // Initialize intermediate result

4      **for** $j \in I_{SL}$ **do**

5          **if** $j \neq 0$ **then**

6              $\widetilde{m}_\Delta \leftarrow \widetilde{m}$ - rowmax($Q_i K_j^T / \sqrt{d}$);    $\widetilde{l} \leftarrow \widetilde{l} \cdot \exp(\widetilde{m}_\Delta)$

7          **end**

8          $\widetilde{m} \leftarrow$ rowmax($Q_i K_j^T / \sqrt{d}$);                 // Ignore causal mask

9          $\widetilde{l} \leftarrow \widetilde{l} +$ rowsum($\exp(\frac{Q_i K_j^T}{\sqrt{d}} - \widetilde{m})$)    $M_{bs}[i, j] \leftarrow 1$

10     **end**

11     **for** $j \leftarrow 1$ *to* $(i - N_{local} - 1)$ **do**

12         $\widetilde{S}_{ij} \leftarrow$ Dequantize($\widetilde{Q}_i \widetilde{K}_j^T$) $/ \sqrt{d}$;          // Approximate attention weight

13         $\widetilde{P}_{ij} \leftarrow \exp(\widetilde{S}_{ij} - \widetilde{m}) / \widetilde{l}$;          // Compute Relative Attention Score

14         $M_{bs}[i, j] \leftarrow \max(\widetilde{P}_{ij}) \geq \tau$

15     **end**

16 **end**

**Output:** Block-level sparse mask $M_{bs}$

---

