# OpenReview forum: "SALE : Low-bit Estimation for Efficient Sparse Attention in Long-context LLM Prefilling"
_ICML.cc/2026/Conference — ICML 2026 regular_

### Official Review · Reviewer_bnNP · 2026-03-08

**Soundness:** 3
**Presentation:** 3
**Significance:** 3
**Originality:** 3
**Overall Recommendation:** 4
**Confidence:** 4

**Summary:**

This paper addresses the computational bottleneck of the self-attention mechanism during the long-context prefilling stage in LLMs. The authors propose SALE, a fine-grained sparse attention method that identifies and retains important query-key pairs while skipping less significant ones. The approach uniquely combines low-bit quantization to estimate attention weights efficiently with a novel Relative Attention Score to accurately assess the importance of query-key pairs, enabling the construction of a highly sparse attention mask. Implemented with a custom CUDA kernel to minimize overhead, SALE achieves significant speedups without requiring parameter training, and it demonstrates superior accuracy-efficiency trade-offs on long-context benchmarks compared to existing sparse attention methods.

**Compliance With Llm Reviewing Policy:**

Affirmed.

**Final Justification:**

My final rating is 4. My concern is solved.

**Key Questions For Authors:**

1. An interesting observation is that the proposed sparse attention sometime outperforms the full attention baseline. While intriguing, this raises a concern regarding the core claim of the paper. If this result is attributed to the variance in the evaluation metrics rather than a genuine algorithmic advantage, it undermines the assertion that the sparse attention method accurately "matches" the full attention baseline. The authors should clarify whether the experiments are controlled to isolate the effect of sparsity from metric noise.

2. See in Limitations.

**Limitations:**

1. A key weakness is the absence of diagnostic experiments that directly motivate SALE's specific design choices. The evaluation treats the method as a black box, demonstrating superior speed-accuracy trade-offs without isolating how components like low-bit estimation or the Relative Attention Score address identifiable failure modes of prior work. Ablation studies that systematically disable these components and measure their impact on specific long-context tasks are needed to verify that each design genuinely targets a distinct bottleneck. Without such analysis, it remains unclear whether the architecture is optimally decomposed or merely over-engineered.

2. The latency reporting aggregates all prefilling attention time, without decomposing standard systems metrics such as time-to-first-token (TTFT), tokens-per-second (TPOT), or interaction with decoding throughput and host–device overheads; this makes it harder to judge user-perceived benefits and pipeline-level trade-offs.

3. The evaluation does not clearly cover scenarios where SALE might *not* be preferable, e.g., shorter contexts, heavily decoding-bound workloads, or settings where calibration cost outweighs the gains.

**Strengths And Weaknesses:**

Strengths

1. A conceptually simple but novel combination of low-bit QK estimation and relative scoring that allows element-wise inspection while avoiding full softmax materialization

2. Strong empirical coverage on three long-context benchmarks and three model families (including an additional Qwen-2.5 in the appendix) with carefully matched baselines and hyperparameter sweeps

3. A solid systems contribution with a two-pass CUDA design, detailed latency breakdowns, and ablations on per-head calibration and quantization schemes.

4. The paper is generally well written, with clear formulations of block-level sparsity, importance approximation, and the calibration process, and it makes a compelling case that SALE improves the speed–accuracy frontier over prior dynamic sparsity approaches

Weaknesses

1. Lack of motivation experiments that driven to the designs

---

> ### Author Rebuttal · Authors · 2026-03-31
>
> We sincerely thank reviewer  for the thorough and constructive review. Below, we address each concern.
>
> ---
>
> **Response to Question**
>
> We agree: these instances reflect evaluation metric variance (e.g., F1 and ROUGE fluctuations), not a genuine advantage of sparsity. The fact that SALE's results fall within this noise band reinforces that its approximation error is small enough to be statistically indistinguishable from metric-level noise. We will revise the wording from "matches" to "achieves comparable performance to" full attention.
>
> ---
>
> **Response to Weakness & Limitation 1**
>
> Thank you for this suggestion. We would like to clarify that SALE proposes a unified framework that leverages holistic and fine-grained inspection to improve block selection accuracy for sparse attention. Some designs within this framework, such as the Relative Attention Score, are not separable modules whose contributions can be measured in isolation, as it is the key component to make on-the-fly block skipping practical. Disabling this design will fail the whole framework.
>
> That said, our paper does provide ablation studies for some individual components: (1) Accuracy evaluation of 4-bit QK full attention (Appendix I), demonstrating that directly using INT4 QK products for attention computation leads to significant accuracy degradation — motivating SALE's use of INT4 as an importance estimator rather than for computation; (2) Sparsity-accuracy tradeoff comparison with SALE w/o QK Quant (Appendix D), demonstrating the feasibility of INT4 draft; (3) accuracy-efficiency trade-off curves validating the necessity of per-head calibration; (4) latency breakdown confirming the effectiveness of our kernel optimization techniques.
>
>
> ---
>
> **Response to Limitation 2**
>
> We have conducted new TTFT measurements on Llama-3.1-8B (single RTX 4090):
>
> | Context Length | 8K | 16K | 32K | 64K | 128K |
> |---|---|---|---|---|---|
> | Full Attention | 1.07s | 2.35s | 5.50s | 14.11s | 41.16s |
> | SALE | 1.04s | 2.16s | 4.47s | 9.63s | 22.11s |
>
> SALE reduces TTFT by up to **46% at 128K** (41.16s → 22.11s). We will include TTFT as a primary metric in the revised paper.
>
> ---
>
> **Response to Limitation 3**
>
> We will add a limitations section covering: (1) short contexts (below ~8K tokens) where sparsity estimation overhead exceeds savings; (2) decoding-bound workloads where prefilling is not the bottleneck. We do not recommend using SALE in these scenarios, so evaluating them would not provide meaningful guidance. We thank the reviewer for this suggestion.

---

> > ### Author Rebuttal · Reviewer_bnNP · 2026-04-03
> >
> > Thanks for the response. I have no more questions, and I will keep my score. Good luck.

---

### Official Review · Reviewer_wbBz · 2026-03-11

**Soundness:** 2
**Presentation:** 3
**Significance:** 3
**Originality:** 2
**Overall Recommendation:** 4
**Confidence:** 3

**Summary:**

This paper proposes SALE, a block-sparse attention method to accelerate long-context LLM inference. The core idea is to use low-bit (4-bit) quantized query-key products to approximate attention weights, followed by Relative Attention Score metric that compares attention weight against the "sink and local" regions to choice important blocks. Blocks with scores below a per-head calibrated threshold are skipped. The method is implemented as a two-pass CUDA kernel: a Selection-Pass (4-bit estimation for block selection) and a Computation-Pass (8-bit sparse attention via SageAttention).

The paper reports experiments on three benchmarks (LongBench, InfiniteBench, RULER) using Llama-3.1-8B-Instruct and Qwen3-4B-Instruct, showing at least 3.36× speedup for sequences longer than 64K tokens with negligible accuracy loss.

The idea of leveraging low-bit Tensor Core instructions for fine-grained attention map estimation is novel and practical. However, the method relies on offline per-head threshold calibration with a very small number of samples, has only been evaluated on a single GPU type (RTX 4090), and the Selection-Pass overhead grows quadratically, which raises scalability concerns for extremely long contexts.

**Compliance With Llm Reviewing Policy:**

Affirmed.

**Final Justification:**

The paper is clear and intuitive. The experiments are strong and have good performance. But I think the overall novelty is slightly incremental. Therefore, I recommend a weak accept.

**Key Questions For Authors:**

1. The error bound θ is set to 0.4 for Llama-3.1 and 4.0 for Qwen-3, a 10× difference. What is the root cause of this discrepancy? Is it attributable to differences in the L1 norm scale of the models' output activations, differences in activation magnitude distributions, or some other model-specific factor? Without a principled explanation, it is unclear how a practitioner should select θ when deploying SALE on a new model. Can the authors propose a normalization scheme (e.g., normalizing θ by the mean output activation norm) that yields a consistent, model-agnostic default value?

2. Calibration is performed on 5 samples drawn exclusively from the Retrieve. KV task—a synthetic needle-in-a-haystack task that may elicit atypically sparse and structured attention patterns. How do the calibrated per-head thresholds τ transfer to tasks with qualitatively different attention distributions, such as multi-hop reasoning (MuSiQue), long-form summarization (GovReport), or code completion? Have the authors conducted an ablation that calibrates on one task category and evaluates on another to measure the generalization gap?


3. The paper's central claim is that SALE performs "holistic and fine-grained" inspection of the attention map, in contrast to the "coarse-grained" approaches of SpargeAttn and FlexPrefill. However, SALE itself operates at a block granularity (bq=64, bk=32), which is the same level at which SpargeAttn constructs its block-level sparse mask. The actual distinction appears to be that SALE uses element-wise low-bit QK products rather than a single representative token per chunk to estimate block importance. Could the authors reframe their contribution more precisely? Specifically, how does the recall rate of SALE's block selection compare to that of SpargeAttn's representative-token approach under identical block sizes and equivalent computational budgets for the estimation step?

**Limitations:**

1. **θ requires per-model tuning but is not acknowledged as a limitation.** The 10× difference between Llama-3.1 (θ=0.4) and Qwen-3 (θ=4.0) is stated without justification, and no selection procedure is provided, directly contradicting the "plug-and-play" claim.

2. **Calibration brittleness is not discussed.** Thresholds are derived from 5 samples of a single synthetic task (Retrieve.KV); the paper provides no analysis of what happens when the inference distribution shifts, nor any guidance on when recalibration is needed.

3. **The "fine-grained" framing is imprecise.** SALE still operates at block granularity (bq=64, bk=32), and accuracy gains over SpargeAttn may partly reflect quantization-induced conservatism rather than genuinely superior estimation. This should be acknowledged to avoid overclaiming.

**Strengths And Weaknesses:**

### Strengths
1. The idea of using low-bit (4-bit) quantized QK products for fine-grained attention map estimation is well-motivated. It exploits the throughput gap between 4-bit and 16-bit Tensor Core instructions (8× on RTX 4090/5090), turning the estimation step into a low-overhead operation (~11% of full attention latency). This is a meaningful contribution over prior coarse-grained methods (representative token-based or subset-based inspection).

2. The Relative Attention Score metric is well-designed. By comparing attention weights against the sink-and-local regions rather than computing full Softmax, the method avoids materializing the entire attention map in DRAM. The mathematical transformation in Eq. (3) that converts the threshold comparison into a single floating-point instruction is an elegant optimization.

3. Comprehensive experimental evaluation. The paper evaluates on three complementary benchmarks (LongBench for general tasks, InfiniteBench for extremely long contexts, RULER for synthetic probing) across two models and one additional model in the appendix (Qwen-2.5-32B). The accuracy-efficiency trade-off curves (Figure 4) provide a convincing comparison against baselines.

### Weaknesses

1. Inconsistent error bound θ across models. The error bound θ = 0.4 for Llama-3.1 and θ = 4.0 for Qwen-3 differ by 10×. The paper provides no principled justification for this large discrepancy. Is there a systematic way to determine θ for a new model? This makes the method less "plug-and-play" than claimed, as θ tuning appears to require non-trivial model-specific adjustment.

2. Offline Calibration Introduces Deployment Complexity and Potential Brittleness.
The per-head threshold τ is calibrated offline on five samples from a single task (Retrieve.KV from InfiniteBench). It is unclear how robust these thresholds are when: (a) the model is used on tasks with very different attention patterns (e.g., code completion vs. document summarization), (b) the model is quantized post-hoc or fine-tuned, or (c) the input distribution shifts significantly. The paper does not provide a systematic analysis of calibration generalization.


3. The "Fine-grained vs. Coarse-grained" Distinction Needs More Rigorous Justification.
The paper's central narrative is that SALE performs "holistic and fine-grained" inspection (Figure 1c) while baselines are "coarse-grained" or "incomprehensive." However, SALE still operates at a block granularity (bq=64, bk=32). The distinction from SpargeAttn (which also uses block-level masks) is not qualitatively one of granularity but rather of the approximation quality of the representative token. This distinction should be stated more precisely to avoid overclaiming.

---

> ### Author Rebuttal · Authors · 2026-03-31
>
> Thank you for the constructive feedback. We address each concern below.
>
> ---
>
> **Response to Weakness 1 & Question 1 & Limitation 1**
>
> We measured the mean per-token L1 norm of attention output at 128K context:
>
> | Model | Mean per-token L1 norm |
> |-------|----------------------|
> | Llama-3.1-8B | 6.20 |
> | Qwen-3-4B | 21.23 |
>
> Qwen-3's output norm is ~3.5× that of Llama-3.1. Since θ bounds the absolute approximation error, models with larger activation scales naturally require larger θ for equivalent relative error tolerance. This explains much of the 10× gap.
>
> More importantly, θ is not sensitive. We checked θ for Qwen-3 (single RTX 4090, latency in ms):
>
> | Context Length | 8K | 16K | 32K | 64K | 128K |
> |---|---|---|---|---|---|
> | SALE θ=2.0 | 81 | 226 | 669 | 2148 | 7336 |
> | SALE θ=4.0 | 75 | 210 | 621 | 1988 | 6764 |
>
> Halving θ from 4.0 to 2.0 increases latency by only ~8%, so coarse setting suffices. It is worth noting that in real deployment, hyperparameter calibration inevitably requires per-scenario tuning, as speed-accuracy requirements vary across use cases. We recommend binary search starting from 0 and 5 to find a suitable θ. Given the low calibration cost (less than 5 minutes for Llama-3.1-8B), this is practical.
>
> ---
>
> **Response to Weakness 2 & Question 2 & Limitation 2**
>
> Our paper already demonstrates calibration robustness across model scales (Qwen-3-4B, Llama-3.1-8B, Qwen-2.5-32B) and 22+ diverse subtasks on 3 benchmarks, all using the same 5-sample KV-retrieval calibration.
>
> To further address this, we conducted a cross-calibration experiment on Llama-3.1-8B, comparing KV-retrieval calibration against natural language calibration (6 samples from MuSiQue, GovReport, RepoBench, which are all included in LongBench):
>
> | Calibration Source | Retrieve.KV | MuSiQue | GovReport | RepoBench | LongBench Sum | Latency (128K) |
> |---|---|---|---|---|---|---|
> | Natural text, θ=0.2 | 43 | 30.85 | 35.23 | 58.23 | 124.31 | 6612 ms |
> | KV-retrieval, θ=0.4 | 61 | 30.10 | 35.45 | 60.75 | 126.3 | 6524 ms |
>
> KV-retrieval calibration generalizes well across all task types, while natural language calibration degrades retrieval performance (Retrieve.KV: 61→43). This is principled: retrieval tasks exhibit highly dynamic attention patterns and impose the strictest sparsity constraints, providing a conservative upper bound on required attention density.
>
> Regarding post-hoc quantization or fine-tuning: calibration takes only several minutes, so we recommend recalibrating after any model modification. We will include this analysis and guidance in the revised paper.
>
> ---
>
> **Response to Weakness 3 & Question 3 & Limitation 3**
>
> We thank the reviewer for this suggestion and will improve our wording to better characterize our method. The core feature of SALE is its holistic and fine-grained inspection of the attention map, aiming to precisely select important blocks. As shown in Appendix E, SALE achieves significantly higher sparsity ratios than SpargeAttn while simultaneously obtaining higher scores across all benchmarks. This demonstrates that our holistic and fine-grained estimation strategy effectively identifies truly important blocks, enabling more aggressive pruning without sacrificing accuracy.

---

> > ### Author Rebuttal · Reviewer_wbBz · 2026-04-02
> >
> > Thanks for your rebuttal, I raise my rate to 4.

---

### Official Review · Reviewer_2GpM · 2026-03-12

**Soundness:** 3
**Presentation:** 3
**Significance:** 2
**Originality:** 2
**Overall Recommendation:** 4
**Confidence:** 5

**Summary:**

This paper mainly tackles the problem of the computation cost in LLM prefill. More specifically, authors propose to use low-bit quantization for approximating the attention weight, and use this weight to compute the block sparse mask for full FA computation. Authors also propose a relative score comparison method for reducing the overhead. Beyond this, authors designed highly-optimized CUDA kernels for SALE. The accuracy of SALE exceeds previous SOTA methods.

**Compliance With Llm Reviewing Policy:**

Affirmed.

**Final Justification:**

Thanks for the response. I have no more questions updated my score.

**Key Questions For Authors:**

Please refer to my comments in the weakness.

**Limitations:**

Authors did not include a limitation discussion.

**Strengths And Weaknesses:**

Strengths:
1. The idea of using low bit quantized attention makes sense to me. It could reduce the approximation error of mean pooling, which is used in SpargeAttention;
2. The design of relative attention can further reduce the compuation overhead;
3. Authors provide end-to-end latency speedup;

Weakness:
1. Lack of comparison with attention quantization methods. Although authors claim that the overhead is about 11%~15% to FA, this does not hold when using FP8 FA. In this case, the overhead of 4-bits attention is non-negligible. Please provide an efficiency comparison in this setting, otherwise the benefits of SALE might diminish in this case.
2. Lack of details about the behavior of different heads. In SALE, the whold block would gets skipped if all tokens's attention weights fall below tau; However, there exisits certain heads where its attention weights tends to diffuse among tokens. Authors did not provide a detail analysis about each head's tau.
3. Authors did not include the performance of SALE on regular benchmarks.
4. Lack of novelty, the idea of attention quantization and sparse attention is not new.

---

> ### Author Rebuttal · Authors · 2026-03-31
>
> Thank for the constructive feedback. We address each concern below.
>
> ---
>
> **Response to Weakness 1.**
>
> We have conducted new experiments comparing SALE against SageAttention-1 (INT8 quantized full attention) on Llama-3.1-8B (single RTX 4090, latency in ms):
>
> | Sequence Length | 8K | 16K | 32K | 64K | 128K |
> |---|---|---|---|---|---|
> | Full Attention (FP16) | 107 | 423 | 1642 | 6404 | 25657 |
> | SageAttention-1 (INT8) | 78 | 285 | 1076 | 3994 | 15592 |
> | SALE (θ=0.4) | 73 | 203 | 592 | 1876 | 6524 |
>
> Although SageAttention already provides significant speedup over FA2, SALE still achieves 1.06×–2.39× speedup over SageAttention, with increasing advantage at longer contexts. Notably, SALE already adopts SageAttention's quantization scheme in its computation pass for the selected blocks. The additional speedup comes entirely from sparsity-based block skipping, which is complementary to attention quantization. We will include this comparison in the revised paper.
>
> ---
>
> **Response to Weakness 2**
>
> The threshold τ is calibrated *independently for each attention head* (Section 3.3), precisely to handle heads with diffuse attention distributions. We computed the Pearson correlation between per-head attention entropy and calibrated τ on Llama-3.1-8B: r = −0.30, confirming that diffuse heads (high entropy) receive smaller τ values, meaning fewer blocks are skipped.
>
> The calibration automatically preserves more context for broadly-distributed heads while aggressively pruning concentrated ones. We will include a more detailed per-head analysis with attention score visualizations in the revised paper.
>
> ---
>
> **Response to Weakness 3**
>
> SALE is designed specifically for long-context prefilling acceleration. Regular benchmarks (e.g., MMLU, HellaSwag) typically involve short inputs that do not exercise long-context prefilling. For short sequences (below ~8K tokens), the overhead of sparsity estimation likely exceeds the savings, so we do not recommend using SALE in such scenarios.
>
> ---
>
> **Response to Weakness 4**
>
> We respectfully disagree that SALE is a straightforward combination of attention quantization and sparse attention.
>
> Quantized attention alone still computes all QK pairs including those contributing negligibly to the output, and directly using ultra-low bit (e.g., 4-bit) for attention computation introduces significant accuracy loss (see Appendix I of our paper), limiting the achievable speedup. Sparse attention skips unimportant QK pairs but struggles to accurately identify which pairs are important, leading to suboptimal accuracy-efficiency trade-offs.
>
> SALE bridges these two lines of work: it uses ultra-low-bit quantization to *guide* sparse attention's QK pair selection, and introduces the Relative Attention Score to enable efficient on-the-fly importance evaluation. This combination achieves the best accuracy-efficiency trade-off among existing approaches, as demonstrated in our experiments.

---

### Official Review · Reviewer_Jojb · 2026-03-12

**Soundness:** 3
**Presentation:** 3
**Significance:** 2
**Originality:** 2
**Overall Recommendation:** 4
**Confidence:** 4

**Summary:**

This paper proposes SALE, a sparse attention framework to accelerate the prefilling stage of long-context LLM inference. The method first computes approximate attention scores using low-bit quantized query–key products, then identifies important query–key pairs based on a relative attention score defined with respect to sink and local tokens. A sparse attention mask is constructed accordingly, and full-precision attention is computed only on the selected entries. The authors also implement a customized kernel that fuses the selection and attention computation for efficiency. Experiments on long-context benchmarks demonstrate favorable accuracy–efficiency trade-offs over existing sparse attention methods.

**Compliance With Llm Reviewing Policy:**

Affirmed.

**Final Justification:**

The proposed method is intuitive and clearly presented, with strong empirical performance. The authors have clarified the differences between their approach and existing work in the rebuttal. However, some comparisons with relevant baselines are still missing, and the analysis of hyperparameters remains insufficiently clear.

**Key Questions For Authors:**

1. How does the proposed method compare with simpler alternatives, such as global Top-K or Top-P selection based on approximate attention scores?
2. How does the proposed method compare with low-bit full attention approaches (e.g., SageAttention) under comparable settings (e.g., latency)?

**Limitations:**

See weaknesses.

**Strengths And Weaknesses:**

### Strengths
1. Improving the efficiency of attention computation is highly relevant for deploying large language models in real-world long-context applications.
2. The proposed method is intuitive and clearly presented, with strong empirical performance. It achieves substantial speedups on long-context benchmarks while maintaining competitive accuracy.
3. The paper provides a detailed description of the CUDA kernel optimizations, offering valuable system-level insights and strengthening the practical impact of the work.

### Weaknesses
1. The idea of using low-bit approximations of attention scores to identify important tokens has been explored in many existing works (e.g., [1–2]). The paper does not clearly explain how the proposed method differs from these approaches, which weakens the overall contribution.

2. The motivation for the Relative Attention Score is not fully convincing. Since the method already computes approximate attention scores for all query–key pairs in low-bit precision, it is unclear why the relative scoring formulation is necessary or how it improves token selection. Moreover, the approach relies on the sink–local attention pattern, while LLMs exhibit more diverse patterns. It remains unclear how the method performs under such cases.

3. The paper mainly compares with sparse attention methods. However, low-bit full attention alone can achieve competitive performance (e.g., SageAttention). Including a baseline that performs full attention with low-bit QK computation but without sparsity would better isolate the benefit of the proposed sparse selection mechanism.

4. The robustness of the calibration procedure in practice is unclear. The method relies on a small amount of data to determine selection hyperparameters (e.g., $\tau$), yet the paper provides limited analysis of how these choices affect performance across different datasets, model scales, or prompts. Additional experiments evaluating the robustness and sensitivity of the calibration process would strengthen the paper.

[1] Liu et al. "Deepseek-v3. 2: Pushing the frontier of open large language models." arXiv preprint arXiv:2512.02556 (2025).
[2] Wang et al. "FIER: Fine-Grained and Efficient KV Cache Retrieval for Long-context LLM Inference." arXiv preprint arXiv:2508.08256 (2025).

---

> ### Author Rebuttal · Authors · 2026-03-31
>
> Thank you for the constructive feedback. We address each concern below.
>
> ---
>
> **Response to Weakness 1:**
>
> We acknowledge that DeepSeek-V3's DSA and FIER share the “low-bit draft + full-precision compute” paradigm with SALE. However, SALE targets a distinct scope: *training-free acceleration of long-context prefilling*. In constrast, DSA requires training a dedicated indexer; FIER targets the decoding phase.
>
> Long-context prefilling poses unique challenges. The attention weight count scales quadratically — at 128K context with 32 heads, a single layer involves ~5.5×10¹¹ weights numbers. Materializing the tensor in GPU memory for Top-K sorting causes OOM and excessive GPU data movement, making the naive “score-then-sort” paradigm impractical.
>
> SALE introduces targeted designs for this setting: (1) Relative Attention Score: evaluates block importance on-the-fly against a sink-and-local reference to avoid O(N²) materialization; (2) custom INT4 CUDA kernel: optimized for tiled, streaming estimation; These designs are specific to the prefilling setting and absent in prior work.
>
> ---
>
> **Response to Question 1:**
>
> As discussed in W1, global Top-K/Top-P requires materializing the full attention score matrix before sorting, which far exceeds GPU memory capacity at long-context scale. This makes such a comparison impractical. SALE's Relative Attention Score sidesteps this bottleneck by evaluating each block independently in a single streaming pass.
>
> ---
>
> **Response to Weakness 2:**
>
> We would like to clarify that: after low-bit QK multiplication, we obtain estimated attention *weights* (logits), not *scores*. Computing true attention scores requires Softmax, which needs the max and sum over *all* key positions — meaning all QK pairs must be computed before any score can be evaluated. As analyzed in W1, storing all these weights in GPU memory is infeasible at long-context scale.
>
> The Relative Attention Score bypasses this limitation. The key insight: if an estimated weight is small compared to the weights from the sink-and-local region (computed in full precision), its post-Softmax score will be even more negligible. Multiple studies [1-3] have shown that sink-and-local regions consistently exhibit large attention weights across most heads, making them a reliable reference. For the minority of heads where sink-local concentration is less pronounced, the per-head calibration procedure automatically lowers the threshold $\tau$, preserving more blocks for those heads.
>
> [1] Xiao et al., "Efficient Streaming Language Models with Attention Sinks", ICLR 2024.
> [2] Gu et al., "When Attention Sink Emerges in Language Models", ICLR 2025.
> [3] Xiao et al., "DuoAttention: Efficient Long-context LLM Inference", ICML 2025.
>
> ---
>
> **Response to Weakness 3 & Question 2:**
>
> We have conducted new latency experiments comparing SALE against SageAttention-1 (INT8 quantized full attention) on Llama-3.1-8B (single RTX 4090, latency in ms):
>
> | Sequence Length | 8K | 16K | 32K | 64K | 128K |
> |---|---|---|---|---|---|
> | Full Attention (FP16) | 107 | 423 | 1642 | 6404 | 25657 |
> | SageAttention-1 (INT8) | 78 | 285 | 1076 | 3994 | 15592 |
> | SALE (θ=0.4) | 73 | 203 | 592 | 1876 | 6524 |
>
> Although SageAttention already provides significant speedup over FA2, SALE still achieves 1.06×–2.39× speedup over SageAttention. Notably, SALE already adopts SageAttention's quantization scheme in its full-precision computation pass. The results above demonstrate that SALE's block skipping provides substantial additional latency reduction on top of attention quantization.
>
> ---
>
> **Response to Weakness 4:**
>
> We would like to clarify that our paper already demonstrates calibration robustness across different model scales (Qwen-3-4B, Llama-3.1-8B, Qwen-2.5-32B) and diverse subtasks on 3 benchmarks, all using the same 5-sample KV-retrieval calibration.
>
> To further address this concern about the calibration dataset choice, we conducted a new cross-calibration experiment on Llama-3.1-8B, comparing our default calibration against calibration on 6 natural language samples from MuSiQue, GovReport, and RepoBench (subtasks of LongBench):
>
> | Calibration Source | Retrieve.KV | MuSiQue | GovReport | RepoBench | LongBench Sum | Latency (128K) |
> |---|---|---|---|---|---|---|
> | Natural text, θ=0.2 | 43 | 30.85 | 35.23 | 58.23 | 124.31 | 6612 ms |
> | KV-retrieval, θ=0.4 | 61 | 30.10 | 35.45 | 60.75 | 126.3 | 6524 ms |
>
> The result shows that KV-retrieval calibration generalizes well across all task types, while natural language calibration degrades retrieval performance. This is principled: retrieval tasks exhibit highly dynamic attention patterns and impose the strictest sparsity constraints, so calibrating on them provides a conservative upper bound. MInference[4] adopts the same calibration principle. We thank the reviewer for this suggestion and will include this analysis in the revised paper.
>
> [4] https://github.com/microsoft/MInference/issues/16

---

> > ### Author Rebuttal · Reviewer_Jojb · 2026-04-02
> >
> > Thank you for the detailed rebuttal and additional experiments. The responses address several of my concerns. That said, a few points could further strengthen the work:
> >
> > 1. Regarding the comparison with Top-K, I believe it is possible to avoid full materialization using streaming or online strategies (e.g., maintaining Top-K indices during tile-wise attention computation). Even setting this aside, including comparisons at shorter sequence lengths where full materialization is feasible would help clarify the relative effectiveness of the proposed selection strategy.
> >
> > 2. The comparison with low-bit full attention is only partially addressed. While the rebuttal includes latency comparisons with SageAttention, the key question is whether sparsity is necessary on top of low-bit approximation. A more informative evaluation would match latency budgets (e.g., using lower-bit full attention) and compare the resulting accuracy trade-offs.
> >
> > 3. The choice of calibration data is especially important in the proposed method, as multiple hyperparameters are determined through calibration. The rebuttal mainly shows that a fixed small number of samples performs well across models and tasks. However, it remains unclear how sensitive the method is to calibration choices. In particular, how performance varies with the number of calibration samples, and whether using more data further improves results, are not analyzed.
> >
> > Given the concerns and limitations discussed, I consider this a borderline paper. Overall, I am slightly inclined to accept due to the strong empirical performance. Addressing the above issues would further improve the clarity and completeness of the paper.

---

> > > ### Author Response · Authors · 2026-04-08
> > >
> > > Thank you for the constructive feedback. We address each point below.
> > >
> > > ---
> > >
> > > ## Concern 1: Comparison with Top-K selection
> > >
> > > Thank you for this insightful question. We carefully considered top-k–based alternatives during the design of SALE and chose not to adopt them for the following reasons.
> > >
> > > Streaming top-k is fundamentally inefficient on modern GPUs. To perform streaming top-k, one must maintain a dynamic data structure such as a min-heap on-chip, tracking the current top-k attention weights as key blocks are processed sequentially. The memory access pattern of this data structure is highly dynamic and cannot be determined at compile time. For example, consider that when a newly computed attention weight happens to be large, it may only need one comparison with the current maximum before insertion; but when it is small, it must be compared against all k maintained entries before being rejected. Since both the access addresses and the number of accesses per key block are data-dependent, the heap data cannot reside in registers — whose access patterns must be statically known to the CUDA compiler — and must instead be placed in shared memory.
> > > Accessing shared memory, however, is far from free: each access incurs tens of clock cycles of latency, and performance degrades further when bank conflicts arise due to the irregular access pattern. As the number of key blocks grows — which is precisely the long-context regime we target — this overhead accumulates and becomes a substantial bottleneck, ultimately negating the speedup that sparse attention is designed to provide.
> > >
> > > As for global top-k, a direct accuracy comparison at meaningful context lengths is unfortunately infeasible. Global top-k requires materializing the entire approximate attention score matrix before selecting entries, which, due to GPU memory constraints, limits evaluation to context lengths of around **8K** tokens before running into OOM. In our experiment, at such short contexts, tasks like KV Retrieval consistently yield perfect scores for all methods, making accuracy comparisons uninformative. We can therefore only compare computational efficiency against global top-k.
> > >
> > > To this end, we developed a dedicated CUDA kernel that writes the INT4-estimated attention weights back to global memory, and then performed softmax, per-block max-pooling, causal masking, and row-wise top-k on the materialized score matrix to obtain the sparse mask. On an L40S GPU with Llama-3.1-8B at **8K** context length, we measured that computing attention for a **single layer** under this global top-k pipeline takes approximately **160 ms**. As a reference, FlashAttention-2 computes attention across all **32** layers in roughly **120 ms** total. The reason for this extreme overhead is that operations such as softmax, top-k, and causal masking must repeatedly load the full attention score matrix (approximately **8 GB** at 8K) from global memory, making memory bandwidth the dominant bottleneck. This confirms that global top-k is not a feasible approach in long-context scenarios, and validate the effectiveness of relative attention score.
> > >
> > > We will include this dicussion in our revised paper.
> > >
> > > ---
> > >
> > > ## Concern 2: Is sparsity necessary on top of low-bit approximation?
> > >
> > > We appreciate this important question. Our paper already provides direct evidence in Appendix B (Table 5) that **INT4 full attention is not a viable alternative**:
> > >
> > > | Method | Retrieve.KV | En.MC |
> > > |--------|------------|-------|
> > > | 4 dense (INT4 full attn, no sparsity) | 41.4 | 60.26 |
> > > | SALE (INT4 estimation + INT8 sparse) | **56.4** | **66.38** |
> > >
> > > Actually，INT4 dense attention suffers severe accuracy degradation (e.g., 15-point drop on Retrieve.KV) even without any pruning, because quantization noise accumulates across all token positions.  As a result, computing accurate attention output requires query and key tensors at no less than 8-bit precision, which limits the speedup achievable through quantization alone. Exploiting sparsity is meaningful.
> > >
> > > --
> > >
> > > ## Concern 3: Sensitivity to the number of calibration samples
> > >
> > > We tested the impact of the number of calibration samples on SALE's performance. The following table reports Retrieve.KV accuracy and latency (in parentheses, ms) at **128K** context length on an L40S GPU, under three sparsity configurations (controlled by the threshold parameter r):
> > >
> > > | Calibration Samples | r=0.4 | r=0.7 | r=1.0 |
> > > |---------------------|-------|-------|-------|
> > > | 20 | **67** (7936) | **39** (6655) | **4** (6407) |
> > > | 5 | **63** (7846) | **41** (6610) | **3** (6383) |
> > >
> > > The results show that increasing the number of calibration samples from 5 to 20 leads to a slight improvement in accuracy, accompanied by a marginal increase in latency. This is expected: more calibration data allows the model to better adapt to the attention patterns present in the Retrieve.KV task, but this also slightly reduces sparsity, resulting in a small latency increase.

---

### Decision · Program_Chairs · 2026-04-30

**Decision:**

Accept (regular)

**Comment:**

This submission proposes a low-bit estimation framework for efficient sparse attention during long-context LLM prefilling. The reviewers found the problem important and appreciated the strong empirical speedups, the optimized CUDA implementation, and the practical efficiency-accuracy trade-off achieved by the proposed relative scoring mechanism. While concerns were raised regarding the incremental nature of the novelty, missing baseline comparisons, and the sensitivity of some calibration choices, the authors’ rebuttal addressed most of these issues. Therefore, I recommend acceptance.